# Fourteen Deaths from Suspected Heparin Overdose in an Italian Primary-Level Hospital

**DOI:** 10.3390/diagnostics13213361

**Published:** 2023-11-01

**Authors:** Nicola Di Fazio, Matteo Scopetti, Giuseppe Delogu, Donato Morena, Alessandro Santurro, Luigi Cipolloni, Gaetano Serviddio, Luigi Papi, Paola Frati, Emanuela Turillazzi, Vittorio Fineschi

**Affiliations:** 1Department of Anatomical, Histological, Forensic and Orthopedic Science, Sapienza University of Rome, 00161 Rome, Italy; nicola.difazio@uniroma1.it (N.D.F.); matteo.scopetti@uniroma1.it (M.S.); giuseppe.delogu@uniroma1.it (G.D.); donato.morena@uniroma1.it (D.M.); asanturro@unisa.it (A.S.); paola.frati@uniroma1.it (P.F.); 2Department of Clinical and Experimental Medicine, University of Foggia, 71100 Foggia, Italy; luigi.cipolloni@unifg.it; 3Department of Medical and Surgical Sciences, University of Foggia, 71100 Foggia, Italy; gaetano.serviddio@unifg.it; 4Department of Legal Medicine, University of Pisa, 56126 Pisa, Italy; luigi.papi@unipi.it (L.P.); emanuela.turillazzi@unipi.it (E.T.)

**Keywords:** healthcare setting, forensic, murder, heparin overdose, fatal hemorrhage, PMCT, autopsy, Glycophorin A

## Abstract

Healthcare-related homicidal cases are not novel within the medical–legal landscape, but investigations are often made difficult with the scarcity of material evidence related to the crime. For this reason, it is necessary to carefully analyze the clinical documentation and employ ancillary forensic resources such as radiology, histopathology, and toxicology. In the presented scenario, the observation of 14 deaths from abnormal bleeding in a First-Level Italian Hospital revealed the administration of massive doses of heparin by a nurse. On behalf of the Judicial Authority, a multidisciplinary medical team investigated the case through the following steps: a thorough review of the clinical documentation, exhumation of the bodies belonging to the deceased patients, performing PMCT and autopsy, and collecting tissue samples for histopathological, immunohistochemical, and toxicological investigations. All the analyzed cases have been characterized by the observation of fatal hemorrhagic episodes not explained with the clinical conditions of the patients, confirmed using autopsy observations and the histological demonstration of the vitality of the lesions. However, due to the limited availability of biological material for the toxicological analysis, the indirect evidence from hematological analyses in hospitalized patients was crucial in demonstrating heparin overdose and its link to the recorded deaths. The present scenario demonstrates the fundamental importance of a multidisciplinary approach to cases of judicial interest related to the healthcare context. Therefore, the illustrated methodologies can be interpreted as an operational framework for similar future cases.

## 1. Introduction

Events such as the execution of murders by healthcare workers in hospitals or other care facilities are not a novelty within the medical–legal field. Recent history is dotted with well-known episodes concerning healthcare-related deaths that are more or less explained with the clinical history of the victims [1]. Following similar events, the term “angel of death”, of Jewish origin [2], has been coined to define a serial killer employed in this specific field [3].

Based on a recent literature review of cases involving health professionals prosecuted or convicted of murder, 60% of these individuals fall into the nursing category, and 18% are in the medical class [4]. However, these data are strongly influenced by the geographical context. Another scientific review focused specifically on the Western context (with 40% of the cases reported in the United States of America) and estimated that nursing staff make up as much as 86% of all killers. Additionally, despite the majority of people in the nursing profession being women, the small percentage of male nursing staff greatly skews this estimate [4].

As for the clinical context in which these crimes are perpetrated, it is essential to state that the preferred categories of patients are fragile individuals suffering from numerous comorbidities, and those of an extremely young or advanced age. Moreover, the hospital environment is the most popular venue for such events. The departments most frequently involved, in accordance with the category of patients targeted by these homicidal gestures, consist of General Medicine, Surgeries, and Intensive Care Units (ICUs) [4]. Within these contexts, access to medicines and drugs is often restricted to nursing staff, so in most cases of murder involving pharmacological substances, the active principle is not subjected to control [5]. For example, the use of insulin, although it is difficult to prove in post-mortem investigations [6].

Other examples of insidious substances include epinephrine, potassium chloride, and possible neuromuscular agents such as succinylcholine and pancuronium, as well as cardiac drugs like digoxin and lidocaine. In cases where the perpetrator does not have authorization to administer drugs, physical means of aggression are more commonly used. Mechanical suffocation through insidious methods, such as forcibly administered water, is a widely employed method [7].

The main problem associated with these events is the difficulty of identifying the cases and initiating judicial investigations, as these acts often go unnoticed due to the fragility of the most frequently targeted patients.

From a statistical perspective, the factor that often raises initial suspicion of the absence of accidental events in relation to numerous deaths is the low probability of a certain number of deaths occurring for the same cause in the same place within a limited time range. Specifically, this is frequently observed as an increase in cardiopulmonary arrests within the same hospital unit. Sometimes, perpetrators can leave a trail behind when changing workplaces, generating a “pseudo-epidemic” of such unusual events in the new setting. During investigations, additional parameters can be challenging to interpret. For instance, in cases of substance intoxication, one may expect an overdose of drugs used in therapy. Post-mortem toxicological investigations, especially after a significant period of time, can be significantly affected by catabolic transformative phenomena that alter the nature and quantity of the substances themselves [8]. However, the fact that non-medical personnel do not have access to these drugs is an important clue in narrowing down the number of suspects.

On the other hand, distinguishing between accidental and intentional administration may be a critical factor in court proceedings. Additionally, in cases where the cause of death is physical violence, the ability to observe and document injuries or wounds is of great assistance to the examiner. Moreover, the presence of multiple types of physical injuries may indicate long-term violence or abuse [9].

Therefore, approaching these cases requires a rigorous methodology and the utilization of all available means based on current scientific knowledge. Sometimes, due to the significant time span between the victims’ death and the initiation of judicial proceedings, traditional forensic techniques may be inapplicable. Consequently, it is often necessary to rely on pre-death blood tests, as biological fluids may be scarce or absent after months or years. However, when a body is buried in a zinc-plated case, it allows for a considerable degree of preservation through special transformative processes such as corification. It is crucial to consider that these processes can lead to post-mortem contamination with compounds like arsenic or lead, due to the chemical interaction with the metal elements of the coffin. Awareness of such phenomena is essential for accurate medical and legal judgment [10,11].

When possible, the approach may involve the exhumation of the body to perform specific clinical, radiological, and laboratory investigations, including the following:post-mortem radiological techniques [12,13,14,15];autopsy investigation;histopathological investigations on samples taken from the deceased;sampling of biological tissues suitable for toxicological investigations [11].

By employing these combined methods, it is possible to reach a correct classification of the cause of death. Integrating these data with the information contained in the clinical records allows for the answering of the specific case’s questions.

## 2. Materials and Methods

Between 2014 and 2015, an Intensive Care Unit of a First-Level Hospital in Italy recorded 14 deaths of patients who, despite suffering from severe morbidity, experienced unexplained hemorrhagic episodes. Due to strong suspicions of criminal activity related to these events, a judicial process was initiated, resulting in the investigation of a nurse for intentional murder.

Forensic investigations were conducted on the clinical records and bodies of eight of these patients, leading to conclusions supported with scientific evidence. These findings provided valuable support for the decision-making process of the Judicial Authority. This paper aims to illustrate the operational methodology employed in these investigations, emphasizing the importance of a multidisciplinary approach in complex forensic cases. The objective of this work is to provide practical and reliable guidance for future investigations in this field.

On the basis of the information provided with clinical and investigative documentation, as well as the analysis relevant for judicial purposes, a multidisciplinary team of experts was appointed by the competent authorities to address the following questions:the administration and dosage of anticoagulant drugs available within the hospital of interest for the 14 patients under investigation;the causes of the 14 deaths and the potential responsibility of the hospital’s health services.

The initial step in the evaluation involved a meticulous examination of the clinical records of the deceased patients. Following this preliminary phase, further forensic investigations were conducted. The selection of the most appropriate investigations for the cases at hand had to consider the advanced state of decomposition of the cadavers.

It is important to note that the judicial autopsies commenced almost a year after the death of the last suspected case (in September 2015), which itself occurred approximately 20 months after the death of the initial case (in January 2014).

The appointed team performed cadaveric exhumation on 8 out of the 14 bodies under consideration (Table 1: Case 1, Case 2, Case 3, Case 4, Case 5, Case 6, Case 10, Case 14). These operations were not possible in the remaining six cases due to objective hindrances including the cremation of the corpse at the time of death.

These bodies underwent initial radiological examinations, specifically total-body Post-Mortem Computed Tomography (PMCT), at the Complex Operational Unit (U.O.C.) of Emergency Radiology at Umberto I General Hospital in Rome. Each corpse was scanned with 16 slices using the following parameters: field of view (FOV) of 50 cm, slice thickness of 0.625 mm, interval of reconstruction of 1.25 mm, 120 kvp, 112 mA, and a 48 s scan time.

After a preliminary radiological assessment, a complete autopsy examination was conducted on the eight bodies at the Morgue of Umberto I General Hospital in Rome. Due to the advanced stage of putrefaction and transformative processes associated with burial in zinc-plated coffins, the visualization of pathological images was hindered. Nonetheless, the investigation allowed for the collection of biological fluids and tissues for further analyses.

Among these investigations, histopathological examinations were performed on the eight bodies, involving the collection of five “standard” samples from the following organs: brain, lung, heart, liver, and kidneys. Additional samples were taken from mucous or cutaneous areas affected by hemorrhagic infiltration.

Two different approaches were employed during these investigations: traditional histochemistry, utilizing van Gieson Elastic and Perls Prussian blue stainings, and immunohistochemistry, involving antigen–antibody reactions for Glycophorin A.

In order to achieve optimal results from the immunohistochemistry investigation, a meticulous preparation methodology was adopted, following the protocol outlined below:pre-treatment of the sample with 0.25 M ethylenediaminetetraacetic acid (EDTA) to facilitate antigenic screening and increase membrane permeability to antibodies;traditional fixation by passing the sample through an alcohol solution and then a formalin solution;washing in water and inclusion in paraffin;production of 4-µm-thick sections from paraffin blocks using a slide microtome;slide mounting by covering with 3-aminopropyl-triethoxysilane;application of anti-Glycophorin antibody A (Santa Cruz, CA, USA) at a concentration of 1:500;incubation of the preparation for 120 min at an ambient temperature of 20 °C;detection using avidin–biotin reaction.

Subsequently, a toxicological investigation was conducted by sampling splenic, hepatic, and renal tissue through enzymatic hydrolysis and generic extraction of acidic and/or basic substances. The employed screening method consisted in liquid chromatography–mass spectrometry (LC-MS) and involved samples processed according to the following steps:solid-phase extraction (SPE) using Isolute HCX columns.cryophylation using nitrogen.inclusion using a 1% solution of acetonitrile and formic acid.untargeted analysis using a liquid chromatography–mass spectrometry device from Thermo.

However, the toxicological tests could not address the fundamental investigative question of whether or not there was an overdose of heparin in the bodies under investigation. The LC-MS method has the ability to detect heparinic derivatives, but the need for large quantities of biological fluids due to losses during extraction made this investigation impracticable. The post-dehydration during autopsy procedures prevented the collection of any fluid useful for the investigation, particularly blood substances.

To partially address this problem, the hematological analyses of the deceased patients, available in all 14 cases, were observed. Specifically, the coagulation tests performed at the time with the Laboratory Analysis of a University Hospital in Italy were considered. Heparinemia was directly measured in the first four cases, providing direct data. In the remaining cases, an indirect evaluation of the biological action of heparin was possible by comparing the values of three parameters: Activated Partial Thromboplastin Time (APTT), Thrombin Time (TT), and Reptilase Time (RT).

In conclusion, it was possible to establish a causal link between the administration of heparin and the recorded deaths.

## 3. Results

A careful analysis of the sequence of events illustrated with the medical records of the 14 cases considered (Table 1) was fundamental as it allowed us to highlight two important aspects related to the case:the major hemorrhagic events could be attributed to the overdose of enoxaparin sodium;the observation of shifts helped establish the relationship between the facts described in the file and the health professionals involved.

Evidence showed that only 12 cases presented a significant hemorrhagic event during hospitalization (cases 1–12), and among these, only 10 cases (cases 1–10) demonstrated a causal link between hemorrhage and death.

Following the initial classification of the cases, a radiological investigation using PMCT was ordered for the available eight bodies. This choice was motivated by the technique’s worldwide success, known for its high sensitivity and specificity in detecting various clinical findings in a minimally invasive manner. The term “Virtopsy”, coined about 20 years ago [16,17], has gained widespread use in the medico-legal field to describe this approach, combining “Virtual” and “Autopsy”.

In the present case, the PMCT technique was particularly suitable as it allowed for the identification of any internal bleeding [18], employing high-energy, low-interference image acquisition protocols [19]. Despite the advanced stages of body transformation, these investigations facilitated a rapid and effective classification of the cases (Figure 1).

Subsequently, to confirm the results obtained from the radiological investigation, complete autopsies were performed on the eight available bodies. The bodies were exhumed by reopening the galvanized coffins in which they were placed. Due to the advanced post-mortem transformative processes, particularly corification, locating biological material (especially fluids) and identifying hemorrhagic skin patterns presented greater difficulty [20]. However, by utilizing the information provided with the clinical documentation, a satisfactory level of accuracy was achieved during the local examination.

Unfortunately, the same results could not be achieved following corpses’ evisceration because of the advanced degree of putrefaction: therefore, for each of them, five “standard” samples were obtained, including the brain, lung tissue, heart, liver, and kidneys. Moreover, when possible, additional samples were taken from mucous or cutaneous areas showing signs of hemorrhagic infarction [21]. Given the highly compromised state of the biological matter, the choice of histopathological methods was influenced. Therefore, the execution of traditional stains like hematoxylin and eosin was postponed due to the scarcity of residual cellular elements, including nuclei, organelles, and cytoplasmic membranes.

Instead, a specific histochemical staining for elastic fibers using van Gieson’s method was employed to achieve satisfactory visualization of structures less affected by post-mortem transformative processes. This staining technique combines multiple reagents (resorcinol–fuchsin for elastic fibers, Weigert’s iron hematoxylin for nuclear staining, picrofuchsin for collagen matrix) to enhance the visualization of residual biological structures that retain significant diagnostic value in forensics [22,23].

Furthermore, histochemical iron staining with Perls Prussian Blue was performed to highlight the presence of hemosiderin deposits [24], and an immunohistochemical study using the antibody reaction against Glycophorin A was conducted in the presented cases [25]. Glycophorin A, an antigen present on the surface of erythrocytes regardless of blood group, was chosen due to its high sensitivity in diagnosing hemorrhagic lesions, even at a considerable post-mortem interval (PMI) when macroscopic examination of tissues is hindered by putrefactive processes [26,27] (Table 2, Figure 2).

During the autopsy, biological tissues from the liver, spleen, and kidney were sampled for toxicological purposes, and the analyses were conducted using LC-MS. This specific choice was made due to the high throughput, soft ionization, and excellent metabolite detection capabilities of LC-MS [28]. Although its diffusion to date is limited (estimated to be less than 1% of modern diagnostic laboratory analyses), the forensic field has shown proficiency in utilizing LC-MS [29].

As previously mentioned, the toxicological investigation aimed to provide a general screening for substances commonly abused, despite the challenges posed with the poor preservation of the tissues under investigation. In all eight cases tested for general acidic or basic substances, the toxicological results were negative.

However, the lack of blood or its scarcity in the investigated cadavers for direct detection of heparinic drugs necessitated the use of indirect methods based on data obtained from the available clinical documentation. In some patients (Cases 1–4), heparinemia measurement during their lifetime provided reliable data. For the remaining cases, a clinical evaluation was conducted based on known values of aPTT, TT, and RT.

These indicators are commonly used in clinical practice to assess blood coagulability and can be influenced by various clinical conditions or drug dosages. aPTT, currently the most widely used method for coagulation screening, detects deficiencies in multiple coagulation factors, including prothrombin (FII) and FX [30]. Since heparin’s biological action involves activating antithrombin (AT), which inhibits the aforementioned factors [31], aPTT serves as a first-level screening method, particularly for Unfractioned Heparins (UFHs) [32]. TT, on the other hand, is useful for detecting qualitative and quantitative fibrinogen abnormalities and is highly sensitive to the presence of heparin [33].

However, the observation of abnormal prolongation of aPTT and TT alone is not a reliable diagnostic element for heparin administration, as several clinical or pharmacological conditions can lead to a similar profile [34]. Therefore, an additional indicator, RT, was evaluated [35]. This test is highly sensitive to fibrinogen abnormalities and is not affected by anticoagulants, including heparin [36].

Exclusion of alternative factors such as oral anticoagulant therapy (OA) was achieved by monitoring the progressive normalization of the International Normalized Ratio (INR) during the hospitalization of patients receiving such therapies for their respective conditions [37].

In this context, the combination of prolonged aPTT and TT without alteration in RT was considered a highly reliable indicator of heparin administration in the investigated cases (Table 3).

The great advantage of this assessment lay in its applicability to all the cases involved in the investigation, rather than just the eight cases for which the bodies were available. Furthermore, the observation of indirect biological effects of heparin overdose at the time of hospital admissions provided scientifically reliable data with a high degree of certainty.

After a comprehensive evaluation of clinical, radiological, autopsy, histopathological, and toxicological data from 14 patients, it was possible to establish the causal link between heparin administration and the recorded deaths (Table 4).

## 4. Discussion

Although clinical documentation highlights hemorrhagic episodes regarding all the cases examined (of which, moreover, only 12 were characterized by clinical severity), the application of rigorous medico-legal methodology has brought to light decisive aspects.

Firstly, the outcome of the described investigations has revealed the actual necessity of careful observation of available medical documentation as the initial approach to each case, whenever it is available.

Subsequently, the exhumation of eight cadavers was the starting phase of all procedures carried out on the considered cases. In fact, it allowed the discovery of subcutaneous hemorrhagic areas in four out of eight cases, which were not only vital (Glycophorin A+) but also consistent with an iatrogenic hemorrhagic diathesis. Unfortunately, due to the poor degree of preservation of the corpses, it was not possible to observe further macroscopic phenomena attributable to bleeding caused by heparin overdose.

Similarly, the application of forensic radiological investigations was not decisive for the detection of hemorrhagic lesions or other elements relevant to the case evaluation. However, the capability of this technique to conservatively visualize numerous findings, including some elements useful for subject identification (e.g., joint or dental prostheses, presence of fracture outcomes, or absence of specific anatomical areas) or for estimating the time of death (liquefaction processes in the parenchyma, adipose degeneration, endovascular gas phenomena), has emerged strongly, confirming what is already known regarding the undeniable usefulness of such techniques for forensic purposes.

Therefore, the results obtained from PMCT in this paper must be explicated with a critical eye. On the one hand, it is crucial to interpret aspects dictated with the progression of putrefactive phenomena that could potentially be misinterpreted as pathological phenomena [38]. On the other hand, the conservative visualization of regions that are difficult to investigate during autopsy (e.g., minor blood vessels) allows for a much higher overall accuracy compared to a simple autopsy. Finally, the fundamental task of PMCT consisted in excluding other modes of death, which is of crucial importance in legal and evidentiary contexts in cases of violent causes [39].

For this reason, it is possible to affirm that the implementation of such a resource within the conducted investigations has not only demonstrated its usefulness but, above all, its standardized applicability in a forensic context.

However, the execution of autopsy investigations has proven to be even more fundamental as it allowed for the objective observation, beyond the mere findings consistent with known pathologies and hospital admissions, of the presence of subcutaneous hemorrhagic extravasations in four out of eight cases, which were not only vital (Glycophorin A+) but also compatible with an iatrogenic hemorrhagic diathesis.

Regarding the demonstration of the vitality of hemorrhagic lesions in decomposed cadavers, immunohistochemical investigation for the detection of Glycophorin is currently considered the gold standard [40,41,42]. In this sense, the present study confirms this assumption by identifying a positive reaction in all investigated areas despite a post-mortem interval (PMI) of over 1 year for all corpses [43,44].

Furthermore, the progression of putrefactive phenomena in the investigated corpses led to the almost total loss of all vital biological fluids (e.g., vitreous humor, cerebrospinal fluid, central and peripheral blood). Therefore, direct access to body cavities was necessary in order to proceed with sampling from solid organs.

Another aspect of significant interest was the interpretation of hematological investigations conducted during the individuals’ lifetime in order to identify laboratory alterations specifically related to heparin administration. Given the fundamental value that such investigations had in the overall inquiry, the present paper emphasizes the crucial importance of a multidisciplinary approach in every forensic case, but particularly in matters related to the healthcare field.

Ultimately, the execution of the presented investigations allowed for a comprehensive response to the questions posed by the judicial authority and has given a solid basis for the conduct of a judicial process that is still ongoing.

## 5. Conclusions

In the forensic field, the hypothesis of homicide perpetrated by healthcare personnel within care facilities constitutes an area of particular difficulty, as often the methods used to carry out the act of suppression are difficult to interpret and not unambiguous. Furthermore, the relative rarity with which such episodes come to light is significantly influenced by the presumed high rate of unreported events to the competent authorities, as they may be interpreted as natural deaths.

However, the advancement of technical–scientific knowledge constantly provides the forensic pathologist and all the physicians involved in the technical operations with new investigative methodologies to arrive at conclusions characterized by a degree of certainty. Specifically, the impossibility to proceed with cadaveric blood heparin dosing was overcome with the indirect finding of APTT and TT elevation with Reptilase Time within physiological limits, allowing the definitive establishment of a causal link between pharmacological overdose and death in 10 patients out of 14 investigated and, therefore, providing valuable data for the final judgement.

Although the scientific literature shows numerous reports on cases not dissimilar to those presented here, the complete and rigorous application of a multidisciplinary and multi-instrumental forensic approach constitutes an experience that has not been previously addressed.

For this reason, the paper’s aim is to demonstrate the effectiveness of a systematic and comprehensive approach to the examination of cases of extraordinary complexity, where a high PMI prevents the implementation of a traditional protocol based solely on autoptical and histopathological investigations [45,46,47].

In conclusion, the investigative methods illustrated here can be considered a true methodological proposal in order to establish an operational pathway, especially for future cases of violent death in the healthcare field [48,49,50,51,52]. “The pursuit of safe care as a new emerging right for patients and balancing the right to legal justice with the right to safer healthcare merit further investigation and discussion” [53].

## Figures and Tables

**Figure 1 diagnostics-13-03361-f001:**
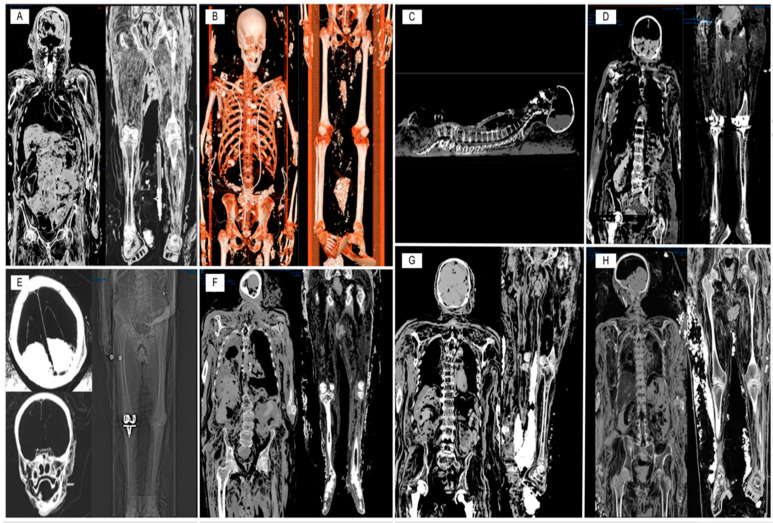
Details of PMCT investigations conducted on 8 exhumed bodies. (**A**) Case 1: widespread putrefactive phenomena characterized by adipose degeneration and gaseous production can be observed. In addition, outcome of surgical treatment for left pertrocanthric fracture is highlighted. (**B**) Case 2: it is possible to observe the progress of putrefactive phenomena, besides the presence of a cardiac pacemaker; a mitral valve prosthesis and lithiasic concretions to the urinary system bilaterally. (**C**) Case 3: a thoracic pacemaker and a fracture of the L3 vertebral body are observed in this case. (**D**) Case 4: the cerebral parenchyma assumes, by virtue of the putrefactive phenomena, the typical radiological aspect of “Swiss cheese”; we also observe the results of the positioning of right hip prostheses and bilateral knee prostheses. (**E**) Case 5: a substantial integrity of the somatic structures is observed, particularly of the skull containing little colliquate brain matter. In detail, result of placement of right total knee prosthesis. (**F**) Case 6: marked colliquative phenomena in fatty tissue. Further acquisitions allowed for objectifying the presence of cardiac pacemaker in situ. (**G**) Case 10: minor putrefactive phenomena, especially in fatty tissue. The spinal canal, however, is occupied by gaseous material of post-mortal origin. Bilateral hip prosthesis outcomes. (**H**) Case 14: evident bodi disintegration consequent to the advance of the putrefactive phenomena. Both kneecaps are dislocated laterally.

**Figure 2 diagnostics-13-03361-f002:**
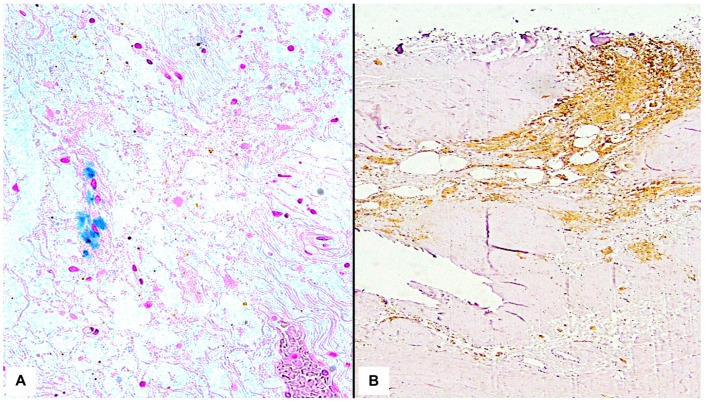
Details of histochemical and immunohistochemical investigations conducted on autoptic biological samples. (**A**) Presence of positive reaction to Perls staining in brainstem. (**B**) Positive reaction to Glycophorin A in encephalic structures.

**Table 1 diagnostics-13-03361-t001:** Summary of clinical data related to the fourteen shown cases.

Case	Age	History	Diagnosis	Clinical Data	Blood Exams	Therapy
Case 1	73	Obesity, arterial hypertension, hypertrophic cardiomegaly.	Trauma from a fall.	**28 September 2015**—9:57 a.m.: The patient arrived at the ED of Piombino Hospital by ambulance due to a fall from approximately 3 m. The patient was in a supine position, with normal hemodynamic state, and symmetric neurological function without focal deficits, but with a noticeable limitation of the left lower limb. Blood chemistry tests revealed hyperglycemia (275 mg/dL) and high-sensitivity troponin of 6.5 pg/mL. The ECG showed “T-wave inversion in precordial leads suggestive of hypertensive heart disease”. X-rays of the cervical spine and left knee were performed, and no fractures were detected. However, an X-ray of the left femur revealed a fragmented and dislocated pertrochanteric fracture. A thoracic–abdominal CT scan was also conducted, which indicated thickening of the pulmonary interstitium, a suspected left adrenal adenoma, and post-traumatic lesions at the L2-L3 level, without any hematomas or post-traumatic vascular injuries. 15:45 p.m.: He was transferred to the local ICU. 18:45 p.m.: He underwent osteosynthesis surgery. 19:45 p.m.: End of surgery. 20:15 p.m.: He returned to the ICU from the operating room still intubated, being monitored, and in spontaneous breath. The surgical wound started bleeding. 20:45 p.m.: An orthopedic consultation was performed, and the wound was dressed with a compression bandage. 21:50 p.m.: He became hypotensive, with cold sweating and agitation. His blood pressure was 50/42, and heart rate was 67. Blood transfusion was performed, but there was new bleeding from the surgical wound. The nursing diary reported “Considerable bleeding from the femoral wound that expands massively; the orthopedist is alerted”. 23:30 p.m.: An orthopedic consultation was performed, and bleeding was controlled from 3 surgical accesses. The compressive dressing was renewed, and a repeat blood transfusion was performed. Intubation was carried out. The patient was soporous and hypotensive with episodes of severe bradycardia, which were later resolved. **29 September 2015**—00:40 a.m.: Severe bradycardia with ST elevation was observed, and eventually, asystole occurred. 01:10 a.m.: Demise.	**28 September 2015** At 9:57 a.m.: HB 14.5 g/dL, RBC 4.79 × 10^12^/L, PLT 179 × 10^9^/L, PT 23%, APTT 106 s, INR 1, Fibrinogen (not available), Antithrombin (not available), D-dimer (not available), Creatinine 0.89 mg/dL. At 09:17 p.m.: HB 8.9 g/dL, RBC 2.95 × 10^12^/L, PLT 179 × 10^9^/L, PT 57%, APTT uncoagulable, INR 1.5, Fibrinogen 266 mg/dL, Antithrombin (not available), D-dimer (not available), Creatinine 1.34 mg/dL. **28 September 2015** At 00:30 a.m.: HB 8 g/dL, RBC 2.67 × 10^12^/L, PLT 160 × 10^9^/L, PT 70%, APTT uncoagulable, INR 1.3, Fibrinogen 222 mg/dL, Antithrombin (not available), D-dimer (not available), Creatinine (not available).	Plasma (1 unit at 11:30 p.m.) Phytomenadione sodium (1 vial at 10:00 a.m.) Tranexamic acid (2 vials at 10:00 p.m.) Prothrombin complex concentrate (1500 IU at 10:00 p.m.) Protamine (1 vial at 00:00 a.m.) Fibrinogen (2 g at 00:00 a.m.) Plasma (1 unit of exchange at 00:30 a.m.)
Case 2	73	Chronic vascular encephalopathy, atrophy of the cerebellar worm, heart dilatation, chronic atrial fibrillation under anticoagulant therapy, hypothyroidism, carrier of cardiac pacemaker and mitral valve prosthesis.	Hypercapnic respiratory failure.	**23 June 2015**—01:41 p.m.: Patient accesses the emergency room. 06:49 p.m.: The patient is transferred to the General Medicine Department due to hypercapnic respiratory failure (type II). Stable clinical status. **23 June 2015**—ENT consultation performed due to absence of swallowing reflex and the need for PEG placement and tracheotomy. **30 June 2015**—00:35 p.m.: Undergoes tracheotomy surgery and PEG placement. Emagel at 500 mL is administered during the procedure. The patient is transferred to the ICU for post-operative monitoring. 04:00 p.m.: Severe bleeding observed from both the tracheotomy and the PEG entry site. 04:30 p.m.: ENT and surgical consultation conducted, and ice applied. Two stitches are placed on the bleeding PEG site, resulting in bleeding cessation. Active bleeding from the tracheostomy continues. 07:10 p.m.: The patient is transferred to the operating room. 08:30 p.m.: Copious bleeding resumes from the tracheotomy site, characterized by bright red blood, which obstructs the airways. 09:15 p.m.: The patient is transferred back to the operating room for tracheotomy revision surgery, revealing abundant bleeding from the bronchi and closure of the tracheotomy. **1 July 2015**—02:45 a.m.: The patient’s condition becomes very serious, experiencing shock and anuria. 05:10 a.m.: Blood pressure measured at 75/45 (heart rate at 45). ECG shows ST segment elevation and negative T-waves. 07:10 a.m.: Blood pressure becomes undetectable, anuria and diffuse marbling observed. ECG shows the presence of PM spikes. Demise occurs.	**23 June 2015** HB 13.6 g/dL, RBC 4.86 × 10^12^/L, PLT 203 × 10^9^/L, PT/, APTT 41 s, INR 2.9, Fibrinogen/, Antithrombin/, D-dimer 102 ng/mL, Creatinine 0.84 mg/dL. **27 June 2015** PLT 144 × 10^9^/L, PT/, APTT/, INR 2.2, Fibrinogen/, Antithrombin/, D-dimer/, Creatinine 0.68 mg/dL. **30 June 2015** At 03:27 p.m.: HB 12.4 g/dL, RBC 4.31 × 10^12^/L, PLT 131 × 10^9^/L, PT 68%, APTT 39 s, INR 1.3, Fibrinogen 449 mg/dL, Antithrombin/, D-dimer/, Creatinine 0.76 mg/dL. At 08:11 p.m.: HB 10.9 g/dL, RBC 3.75 × 10^12^/L, PLT 118 × 10^9^/L, Repeat PT, Repeat APTT, Repeat INR, Repeat Fibrinogen, Antithrombin/, Repeat D-dimer, Creatinine/. At 10:07 p.m.: Uncoagulable PT, Uncoagulable APTT, INR/, Fibrinogen 103 mg/dL, Antithrombin/, D-dimer 113 ng/mL, Creatinine/. At 11:24 p.m.: HB 8.6 g/dL, RBC 3.02 × 10^12^/L, PLT 109 × 10^9^/L, Uncoagulable PT, Uncoagulable APTT, INR/, Fibrinogen 100 mg/dL, Antithrombin/, D-dimer 91 ng/mL, Creatinine 0.74 mg/dL. **1 July 2015** 03:20 a.m.: HB 6.7 g/dL, RBC 2.39 × 10^12^/L, PLT 167 × 10^9^/L, PT 57%, APTT Uncoagulable, INR 1.5, Fibrinogen 251 mg/dL, Antithrombin 61%, D-dimer 83 ng/mL, Creatinine/.	**23 June 2015** Warfarin (½ tablet per day) **26 June 2015** Enoxaparin sodium (6000 IU per day) **27 June 2015** Enoxaparin sodium (6000 IU ×2 per day) **30 June 2015** Enoxaparin sodium (6000 IU) Phytomenadione sodium (1 vial) Prothrombin complex concentrate (500 IU at 20:30 + 2000 IU at 09:15 p.m.) Tranexamic acid (500, 1 vial through OT tube) + Prothrombin complex concentrate (2000 IU, 500 mL + 500 mL during the second revision surgery) Plasma (1 unit at 10:50 p.m.) **1 July 2015** Prothrombin complex concentrate (3500 IU at 01:00 a.m.) Plasma (1 unit at 02:00 a.m.) Fibrinogen (2 g at 02:45 a.m.) RBCs (1 unit at 05:10 a.m.)
Case 3	82	Heart dilation, with a pacemaker and mitral valve prosthesis, moderate aortic insufficiency, chronic atrial fibrillation under anticoagulant therapy, and chronic pancreatitis.	Heart failure.	**07 March 2015**—09:46 a.m.: Admission to Piombino Hospital for heart failure and acute bronchitis. 03:00 p.m.: Transferred to the Department of General Medicine for heart failure. Clinically stable. **10 March 2015**—Urgent thoracoabdominal CT scan performed for respiratory failure and dyspnea during aerosol therapy, revealing ‘severe interstitial pneumonia with thrombosis of the portal branch and splenic infarction, with no signs of pulmonary embolism.’ 08:10 p.m.: Patient transferred to the local ICU due to acute respiratory failure in a cardiac patient, with thrombosis in the hepatic artery and splenic infarction. **11 March 2015**—03:30 a.m.: Hypotension with oliguria, accompanied with psychomotor agitation and confusion. 06:20 a.m.: Septic shock, with severe conditions and decreased blood pressure. Urinary catheter inserted, revealing the presence of hematuria. 08:30 a.m.: Severe clinical conditions persist (hypotension and anuria) with unstable hemodynamics despite aminergic support, and extensive hematomas in the inguinal area. 04:00 p.m.: Pediatric central venous catheter (CVC) placed in the right arm brachial vein, followed by rapid infusion of 500 mL of 5% Albumin for blood pressure of 47/30. Anuria persists. 05:00 p.m.: Respiratory arrest with pulseless electrical activity (PEA). Resuscitation–ventilation and administration of adrenaline performed without success. 05:10 p.m.: Patient’s passing.	**07 March 2015** HB 15.2 g/dL, RBC 4.85 × 10^12^/L, PLT 67 × 10^9^/L, PT/, APTT 47 s, INR 7.6, Fibrinogen 381 mg/dL, Antithrombin 49%, D-dimer 609 ng/mL, Creatinine/. **08 March 2015** APTT 59 s, INR 5.6, Fibrinogen/, Antithrombin/, D-dimer 102 ng/mL, Creatinine 0.84 mg/dL. **10 March 2015** At 07:00 a.m.: HB/, RBC/, PLT/, PT 40%, APTT 42 s, INR 1.9, Fibrinogen 426 mg/dL, Antithrombin 47%, D-dimer 1180 ng/mL, Creatinine/. At 08:48 a.m.: HB 15.1 g/dL, RBC 4.78 × 10^12^/L, PLT 27 × 10^9^/L, PT/, APTT/, INR/, Fibrinogen/, Antithrombin/, D-dimer/, Creatinine/. At 08:29 p.m.: HB 13.7 g/dL, RBC 4.36 × 10^12^/L, PLT 22 × 10^9^/L, PT repeat, APTT repeat, INR repeat, Fibrinogen repeat, Antithrombin repeat, D-dimer repeat, Creatinine 1.30 mg/dL. At 11:18 p.m.: HB/, RBC/, PLT/, PT/, APTT > 400 s, INR > 15, Fibrinogen/, Antithrombin 32%, D-dimer 1185 ng/mL, Creatinine/. **11 March 2015** At 07:00 a.m.: HB 7.9 g/dL, RBC 2.53 × 10^12^/L, PLT 42 × 10^9^/L, PT uncoagulable, APTT uncoagulable, INR/, Fibrinogen/, Antithrombin uncoagulable, D-dimer/, Creatinine 1.74 mg/dL. At 03:48 p.m.: HB 6.3 g/dL, RBC 2.06 × 10^12^/L, PLT 24 × 10^9^/L, PT/, APTT 34 s, INR 2.1, Fibrinogen 216 mg/dL, Antithrombin/, D-dimer/, Creatinine 2.32 mg/dL.	**07 March 2015** Phytomenadione sodium (1/3 vial orally) **08 March 2015** Phytomenadione sodium (1 vial orally) **10 March 2015** Phytomenadione sodium (1/3 vial orally) Antithrombin III (at 11:00 p.m.) **11 March 2015** PLT pool (at 01:30 a.m.) Plasmapheresis Phytomenadione sodium (1 vial at 01:00 p.m.)
Case 4	77	Hip and knee implants, non-Hodgkin’s lymphoma, former smoker.	Heart failure, atrial fibrillation.	**2 January 2015**—05:18 p.m.: Access to the ED for intense asthenia and acute respiratory failure (pH 7.43, pCO_2_ 52 mmHg, pO_2_ 46 mmHg, SO_2_ 88.4%, HCO_3_ 34.5 mmol/l). 08:20 p.m.: Patient transferred to the Department of Cardiology for low-flow heart failure, dyspnea, and hyperpyrexia, associated with unknown onset atrial fibrillation and elevated troponin I. Clinical condition stable, and asymptomatic for angina. Echocardiogram shows left ventricular overload with concentric wall hypertrophy, EF 60%, and mild aortic insufficiency. ECG shows atrial fibrillation. CT scan shows alveolitis. Blood tests reveal progressively decreasing troponin I, decreasing BNP, lymphocytic leukocytosis, and elevated CRP values. Arterial blood gas analysis shows hypercapnic respiratory failure. **4 January 2015**—12:00 a.m.: Patient transferred to the local ICU for sepsis and acute respiratory failure in non-Hodgkin’s lymphoma. 07:00 p.m.: Patient becomes increasingly drowsy, and endotracheal intubation is performed after sedation and preparation. **6 January 2015**—Afternoon: Peripheral venous access is removed, and a right femoral CVC (central venous catheter) is placed. From the CVC management monitoring board: “Heparin for flushing: No”. **7 January 2015**—12:00 a.m.: Hematological consultation reveals the current hematological situation is not concerning and will be reassessed if there is an improvement in clinical conditions. Night shift: CVC dressing changed due to excessive bleeding. Patient weighed. **8 January 2015**—08:00 p.m.: Presence of hematuria. Night shift: Blood clot aspirated from the endotracheal tube, and hematuria observed in the evening. Coagulation test results were abnormal, with ATIII within normal range. Coagulation test repeated, confirming abnormal coagulation. Bright red blood aspirated. Patient presents hematuria and bleeding from the mouth during aspiration. **9 January 2015**—06:00 a.m.: Presence of blood from the NG tube. Aspirated clots and performed wash. The patient opens her eyes and responds to simple commands. Medication administered at the femoral insertion site. 08:00 a.m.: Stable parameters, but bleeding continues from the NG tube. 08:30 a.m.: Patient in critical condition, with bleeding from the NG tube, mouth, and hematoma at the femoral vein site. 01:45 p.m.: Worsening hemodynamics despite fluid resuscitation. 02:15 p.m.: Demise.	**4 January 2015** At 07:00 a.m.: HB 13.8 g/dL, RBC 4.68 × 10^12^/L, PLT 308 × 10^9^/L. **5 January 2015** At 08:32 a.m.: HB 12.6 g/dL, RBC 4.22 × 10^12^/L, PLT 258 × 10^9^/L, PT 83%, APTT 23 s, INR 1.1, Fibrinogen 464 mg/dL, Antithrombin/, D-dimer/, Creatinine 1.74 mg/dL. **9 January 2015** At 07:00 a.m.: HB 10.7 g/dL, RBC 3.59 × 10^12^/L, PLT 247 × 10^9^/L, PT/, APTT/, INR/, Fibrinogen/, Antithrombin/, D-dimer/. At 10:38 a.m.: HB 10.1 g/dL, RBC 3.43 × 10^12^/L, PLT 258 × 10^9^/L, PT 63%, APTT uncoagulable, TT > 300 s, INR 1.3, Fibrinogen 352 mg/dL, Antithrombin 87%, D-dimer 3162 ng/mL. At 11:01 a.m.: HB 9.9 g/dL, RBC 3.37 × 10^12^/L, PLT 288 × 10^9^/L, PT/, APTT/, INR/, Fibrinogen/, Antithrombin/, D-dimer/.	**3 January 2015** ASA (100 mg per day) Fondaparinux sodium (2.5 mg per day) **5 January 2015** Fondaparinux sodium 2.5 mg per day **8 January 2015** Vitamin K (1 vial in 100 cc normal saline) **9 January 2015** Plasma (5 units + 3 units)
Case 5	88	Severe cerebrovascular and degenerative dementia with psychosis, diabetes mellitus, chronic atrial fibrillation under Coumadin therapy, chronic HCV and HBV infection.	Chronic subdural hematoma with recent bleeding.	**25 November 2014**—11:42 a.m.: Access to the ED for recent facial trauma (3 days prior). Neurological impairment observed (dysphasia, disorientation, GCS 14). A cranial CT scan is performed, revealing chronic subdural hematoma (CSH) with recent bilateral temporal-fronto-parietal bleeding. The patient is transferred to the local ICU. 06:00 p.m.: Epistaxis occurs. 06:50 p.m.: The patient undergoes a head CT control after 6 h, which shows no change. 09:00 p.m.: INR 1.4. A blood sample is sent to determine the presence of coagulopathy (indeterminable). Even after repeating the test, the sample still yields an indeterminable result (the laboratory mentions a ‘strange’ coagulation curve that cannot be interpreted)… The patient tends to experience bleeding from venous accesses and the oral cavity. **26 November 2014**—06:30 a.m.: Neurological condition worsens, with the patient entering a state of coma. Midriatic pupils. GCS 3/15. INR 3.6. 08:55 a.m.: A head CT scan reveals an increase in the thickness of the subdural hematoma (maximum thickness of 3 cm), temporal herniation with compression of the midbrain, and a sickle hernia with midline deviation of 1.5 cm. 01:10 p.m.: The patient passes away.	**25 November 2014** At 12:08 a.m.: Hb 11 g/dL, RBC 3.58 × 10^12^/L, PLT 247 × 10^9^/L, PT not available, APTT 34 s, INR 2.5, Fibrinogen level not available, Antithrombin level not available, D-dimer level not available, Creatinine level 0.72 mg/dL. At 11:32 p.m.: HB not available, RBC not available, PLT count, PT undetermined, APTT not available, TT undetermined. **26 November 2014** At 06:25 a.m.: HB 8.9 g/dL, RBC 2.83 × 10^12^/L, PLT 262 × 10^9^/L, PT not available, APTT uncoagulable, INR 3.9, Fibrinogen 686 ng/dL, Antithrombin level not available, D-dimer level not available, Creatinine 0.69 mg/dL.	**25 November 2014** Tranexanic acid + Phytomenadione sodium (1 vial at 6:00 p.m.) **26 November 2014** Prothrombin complex concentrate (1000 IU)
Case 6	88	Bronchial asthma, ankylosis of the left upper limb, diabetes mellitus type 2, paroxysmal advanced atrioventricular block with recent implantation of bicameral pacemaker and VDD (single-lead atrial sensing ventricular pacing).	Pulmonary embolism with hemodynamic instability, accompanied with sepsis likely caused by a pacemaker infection.	**19 June 2014**—Accessed the ED of another hospital due to disorientation and agitation. Underwent head CT, ECO-cardio, venous ecodoppler of the upper left limb, and pulmonary angio-CT, which showed the absence of hemorrhagic or ischemic brain lesions, presence of a thrombus in the right atrium, complete recent thrombosis of the left subclavian vein and the ipsilateral internal jugular, and pulmonary embolism. **24 June 2014**—11:30 p.m.: Transferred to the ICU of Piombino Hospital due to hemodynamic instability caused by pulmonary embolism (PE). **25 June 2014**—Morning (7:00 a.m./02:00 p.m.): Serious general condition … Underwent blood sampling for aPTT control every 4 h starting from 10:00 a.m. (02:00 p.m.–06:00 p.m.–10:00 p.m.–02:00 a.m.–06:00 a.m.)”. Underwent urination, rectal swab, and sputum tests, which came back negative. Blood culture tested positive for Staph. Epidermidis. Night (09:00 p.m./07:00 a.m.): Blood withdrawal performed at 22:00 and 24:00, but it was not possible to evaluate it. Subsequent samplings were performed every 6 h instead of every 2 h… **26 June 2014**—09:30 a.m.: Serious clinical condition with signs of bleeding from venous access. Heparin was suspended. 02:40 p.m.: Serious clinical condition. Anuria. 11:00 p.m.: Patient remains hypotensive and anuric, presence of digested blood in the bag connected to the NG tube. 27.06.2014—07:50 a.m.: Patient passed away.	**24 June 2014** At 11:45 p.m.: Hb 13 g/dL, RBC 4.24 × 10^12^/L, PLT 147 × 10^9^/L, PT 59%, APTT 38 s, INR 1.4, Fibrinogen 568 mg/dL, Antithrombin not available, D-dimer 4117 ng/mL, Creatinine 1.05 mg/dL. **25 June 2014** At 10:00 a.m.: HB (not available), RBC (not available), PLT (not available), PT 53%, APTT 115 s, INR 1.5, Fibrinogen 519 ng/dL, Antithrombin (not available), D-dimer (not available), Creatinine (not available). At 05:43 p.m.: HB 11.1 g/dL, RBC 3.7 × 10^12^/L, PLT 141 × 10^9^/L, Repeated PT, Repeated APTT, Repeated INR, Repeated Fibrinogen, Antithrombin (not available), D-dimer (not available), Creatinine 0.7 mg/dL. **26 June 2014** At 06:00 a.m.: HB 8.2 g/dL, RBC 2.77 × 10^12^/L, PLT 181 × 10^9^/L, PT not executable, APTT not executable, INR not executable, Fibrinogen not executable, Antithrombin 51%, D-dimer (not available), Creatinine 0.78 mg/dL. At 10:54 a.m.: PT 39%, APTT > 240 s, INR 1.9, Fibrinogen 186 ng/dL, Antithrombin (not available), D-dimer (not available), Creatinine (not available). At 11:06 p.m.: PT (not available), APTT 93 s, INR 2.1, Fibrinogen 327 ng/dL, Antithrombin (not available), D-dimer (not available), Creatinine (not available). **27 June 2014** At 07:00 a.m.: HB 5.1 g/dL, RBC 1.71 × 10^12^/L, PLT 167 × 10^9^/L, PT 39%, APTT 49 s, INR 1.9, Fibrinogen 312 mg/dL, Antithrombin 51%, D-dimer (not available), Creatinine 1.90 mg/dL.	**24 June 2014** Ringer lactate (83 mL/h at 11:00 p.m.) Midazolam (45 mg at 22:30) Omeprazole (40 mg) + Levofloxacin (500 mg at 11:00 p.m.) Heparin sodium (25,000 IU and Vancomycin (1 g at 00:00 a.m.) **25 June 2014** Poligeline (500 mg at 07:30 a.m.) Metronidazole (500 mg) and Ranitidine (1 tablet). Discontinue Ringer lactate and Heparin sodium. Calcium heparin (20,000 IU), Midazolam (50 mg), and Furosemide (60 mg). Glucose and saline solution (5%) Metronidazole (500 mg) Oxacilline sodium monohidrated (3 g, 20%), Albumin and Rifampicin (600 mg), Gentamycin (80 mg), and Metronidazole (500 mg). Oxacilline sodium monohidrated (3 g) **26 June 2014** Oxacillin (3 g) Metronidazole (500 mg) Gentamycin (80 mg) Ranitidine (1 tablet), Albumin 20% Glucose 5% Saline solution. Midazolam (50 mg) and Furosemide (80 mg) Isolyte (2000 mL), Oxacillin (3 g), Gentamycin (80 mg) Metronidazole (500 mg) Lispro insulin. 20% Albumin Oxacillin (3 g) Gentamycin (80 mg), Metronidazole (500 mg)
Case 7	88	Chronic cerebral vasculopathy, previous osteosynthesis of the right femur under ASA therapy.	Fracture in the right humeral meta-diaphysis.	**15 December 2014**—11:58 a.m.: Admitted to the ED for suspected fracture of the right humerus. Presents extensive hematoma in the arm and right shoulder with pain, limited mobility, and joint crepitus during movement. Appears uncooperative and disoriented, without signs of neurological deficits. Chest, spine, rib cage, right shoulder, and arm X-rays are performed, revealing a comminuted displaced meta-diaphyseal fracture of the proximal right humerus. Abdominal ultrasound shows no traumatic injuries or abdominal effusions. Brain and spine CT scan demonstrate no intra- or extra-parenchymal hematomas or focal abnormalities in the subtentorial region. 04:36 p.m.: Transferred to the Orthopedics and Traumatology department for the comminuted diaphyseal fracture of the right humerus. 08:00 p.m.: Nursing diary reports bleeding from the bandage. **16 December 2014**—08:00 a.m.: Clinical diary reports blood loss from the Desault brace. Upon opening the bandage, a full-thickness skin lesion is observed and treated with a new dressing. 07:30 p.m.: Nursing diary reports serous fluid discharge from the wound. Internal medicine consultation is conducted, and the patient is alert, calm, not dyspneic or tachypneic, and experiencing severe pain during mobilization attempts, without added sounds, tachycardia, or dependent edema. **17 December 2014**—10:50 a.m.: Anesthesia consultation is performed for evaluation before orthopedic surgery (ASA IV). 07:10 p.m.: Transferred to the Intensive Care and Resuscitation unit for monitoring during a high-risk transfusion. **18 December 2014**—The clinical diary reports stable vital signs. CT scans of the right and left shoulder are performed, revealing a comminuted metaepiphyseal fracture of the right humerus with diastatic fragments and an impacted metaepiphyseal fracture of the left humerus with upward displacement of the bone fragment. **19 December 2014**—09:30 a.m.: Orthopedic consultation is conducted, which does not indicate surgery for the fracture. Sedation is recommended for the placement of a brace in 45° adduction position and suturing of the exposure on the left arm (to minimize the risk of bleeding). 07:00 p.m.: Orthopedic consultation is performed due to bleeding from the fracture, which started approximately 30 min earlier. It reveals an exposed bone stump with bleeding from the fracture site, without arterial and/or venous bleeding. Urgent “open” stabilization surgery is recommended. The patient undergoes open reduction and internal fixation of the right humerus. The fracture is reduced, and osteosynthesis is performed using 3 cannulated screws. 11:23 p.m.: Orthopedic consultation is conducted, and the dressing is changed due to bleeding from the surgical wound. **20 December 2014**—From the clinical diary: “Severe hemorrhagic shock with coagulation refractory to therapy”. 03:30 a.m.: The patient is pronounced deceased.	**17 December 2014** At 07:00 a.m.: HB 8.9 g/dL, RBC 3.11 × 10^12^/L, PLT 208 × 10^9^/L, PT 89%, APTT 18 s, INR 1.1, Fibrinogen 751 mg/dL, Antithrombin/. **18 December 2014** At 07:00 a.m.: HB 9.4 g/dL, RBC 3.34 × 10^12^/L, PLT 248 × 10^9^/L, PT 88%, APTT 25 s, INR 1.1, Fibrinogen 648 mg/dL, Antithrombin/. **19 December 2014** At 07:12 p.m.: HB 10.2 g/dL, RBC 3.57 × 10^12^/L, PLT 266 × 10^9^/L. At 11:51 p.m.: HB 5.5 g/dL, RBC 1.98 × 10^12^/L, PLT 202 × 10^9^/L, PT undetermined, APTT undetermined, INR uncoagulable, Fibrinogen/, Antithrombin 45%. **20 December 2014** At 02:55 a.m.: HB 8.5 g/dL, RBC 2.86 × 10^12^/L, PLT 120 × 10^9^/L, PT/, APTT undetermined, INR 6.4, Fibrinogen undetermined, Antithrombin/.	**15 December 2014** Ivor 3500 **17 December 2014** Bemiparin sodium (3500 at 8:00 p.m.) Packed RBCs 1 U **19 December 2014** Packed RBCs 1 U, Plasma 1 U, antithrombin III 1000 U Vitamin K (1 vial), Prothrombin complex concentrate 1000 U **20 December 2014** Packed RBCs 1 U, Plasma 1 U
Case 8	76	Arterial hypertension, Parkinson’s disease, thoracic aortic aneurysm, previous prostatectomy, depressive syndrome, hepatic steatosis and psoriatic arthritis, reported diagnostic assessment for pulmonary neoformation.	Wheezing.	**26 September 2014**—09:08 p.m.: Presented at the ED with worsening dyspnea (pH 7.47, pCO2 36 mmHg, pO2 62 mmHg, pO2 95.5%, HCO3 26.2 mmol/l, P/F 295 mmHg, Lac 1.6 mmol/L). The patient had fever, productive cough, and marked neutrophilia (84% neutrophils). On physical examination, the patient was alert but intensely dyspneic, with coarse crackles throughout the lung fields, expiratory wheezing, and diffuse rhonchi. Chest X-ray showed decreased transparency in the mid-right field suggestive of consolidation. **27 September 2014**—09:20 a.m.: Transferred to the General Medicine ward due to respiratory failure with hyperpyrexia and right lower lung consolidation. The patient was awake, alert, and dyspneic during speech. Serial blood gas analysis showed hypercapnic respiratory failure with respiratory acidosis (pH 7.28, pCO_2_ 65 mmHg, pO2 75 mmHg). 28/09 01:55 pH 7.28, pCO_2_ 65 mmHg, pO_2_ 83 mmHg. **28 September 2014**—02:30 a.m.: Transferred to the ICU due to respiratory failure caused by right-sided pneumonia. **29 September 2014**—The patient’s condition is severe. Sedation and neuromuscular blockade were initiated due to poor patient–ventilator synchrony. Chest CT scan revealed consolidation in the upper left and upper right lobes. **30 September 2014**—12:00 a.m.: The patient is sedated with stable vital signs. ASA medication was discontinued due to epistaxis. Bronchoscopy was performed, and abundant secretions were suctioned from the right upper lobe bronchus, followed by bronchial lavage. **1 October 2014**—Sudden onset of significant nasal–oral bleeding with aspiration of fresh blood from the nasal, oral, and pharyngeal regions. 04:00 p.m.: ENT consultation reveals epistaxis. 05:00 p.m.: Profuse epistaxis and bleeding from the mouth. Anterior nasal packing followed by posterior packing is performed. 06:00 p.m.: ENT consultation with posterior packing. 07:10 p.m.: Suspected DIC. 09:07 p.m.: Nursing record: “Patient on intermittent positive pressure ventilation (IPPV). Critical condition. Bleeding from the mouth. Fever. 00:00 a.m.: Expectorated sputum culture report (common bacteria and fungi) shows negative results. **2 October 2014**—06:55 a.m.: Clinical record—continued significant blood loss. Extremely critical clinical condition. 07:10 a.m.: Pronounced deceased.	**28 September 2014** At 07:00 a.m.: HB 13.6 g/dL, RBC 4.51 × 10^12^/L, PLT 144 × 10^9^/L, PT 66%, APTT 26 s, INR 1.3, Fibrinogen 1029 mg/dL, D-dimer 2232, Creatinine 1.24, Antithrombin/. **01 October 2014** At 07:00 a.m.: HB 12.2 g/dL, RBC 4.13 × 10^12^/L, PLT 174 × 10^9^/L, PT 83%, APTT 24 s, INR/, Fibrinogen 885 mg/dL, Antithrombin/. At 05:28 p.m.: HB 11.6 g/dL, RBC 3.91 × 10^12^/L, PLT 142 × 10^9^/L, PT/, APTT uncoagulable, INR/, Fibrinogen/, Antithrombin 95%. At 09:36 p.m.: HB 11.1 g/dL, RBC 3.69 × 10^12^/L, PLT 145 × 10^9^/L, PT uncoagulable, APTT uncoagulable, INR/, Fibrinogen uncoagulable, Antithrombin 100%. **02 October 2014** At 07:00 a.m.: HB 7.8 g/dL, RBC 2.65 × 10^12^/L, PLT 224 × 10^9^/L, PT/, APTT uncoagulable, INR 4.0, Fibrinogen 582 mg/dL, Antithrombin/.	
Case 9	77	Scleroderma, depressive syndrome, dyshyroidism and venous insufficiency in the lower limbs.	Pulmonary embolism.	**11 September 2014**—08:11 a.m.: Admitted to the ED due to fever and dyspnea for approximately 10 days. On physical examination, there is a rapid and rhythmic cardiac action, no added heart sounds, reduced breath sounds in the chest, no signs of peripheral vascular disease, and no dependent edema. Chest X-ray reveals mild inflammatory interstitial alveolar opacities in the left basal region and slight indications of chronic obstructive pulmonary disease (COPD). Chest CT scan shows the presence of embolic material in the segmental branches directed towards the medial segments and lower lobes of the right lung, consolidation in the lung parenchyma, and pericardial effusion. 01:40 p.m.: Transferred to the Cardiology Department with a diagnosis of pulmonary embolism. Echocardiography shows aortic regurgitation, mitral regurgitation, and an ejection fraction (EF) of 60%, along with tricuspid regurgitation. During hospitalization, the patient remains stable. Reports dyspnea, slight tachycardia, and hyperpyrexia. **18 September 2014**—04:45 p.m.: Transferred to the ICU with a diagnosis of acute respiratory failure. According to the Clinical and Nursing records, the patient’s vital signs are stable, and there is no fever. **19 September 2014**—07:00 a.m.: Negative urine culture. 08:30 a.m.: the patient is dyspneic but breathing spontaneously. A chest CT scan reveals worsening bilateral parenchymal involvement characterized by multiple diffuse areas of bilateral parenchymal consolidation, no pleural effusion, and no signs of pulmonary embolism. BiPap non-invasive ventilation is applied. According to the Clinical and Nursing records, the patient’s vital signs are stable, and there is no fever. **20 September 2014**—the patient is cooperative, with stable hemodynamics and preserved diuresis. According to the Nursing records, the patient is occasionally agitated and confused. **21 September 2014**—03:35 p.m.: Cardiorespiratory arrest, chest compressions, and oro-tracheal intubation. Initiation of mechanical ventilation, critical clinical condition. Presence of hematuria. 07:00 p.m.: there is significant bleeding from the mouth and the tube. 07:45 p.m.: bright red blood is observed from the oro-tracheal tube and the tube. 12:00 a.m.: According to the clinical diary— bradycardia (34 bpm), gasping, midriatic non-reactive pupils. Bright red blood is aspirated from the trachea and mouth, and hematuria appears. Pharmacologically refractory cardiac arrest. 01:12 a.m.: Pronouncement of death.	**11 September 2014** At 08:32 a.m.: Hb 11.5 g/dL, RBC 4.6 × 10^12^/L, PLT 283 × 10^9^/L, PT 77%, APTT 22 s, INR 1.2, Fibrinogen 542 mg/dL, D-dimer 2537, Creatinine 1.25, Antithrombin/. **17 September 2014** At 07:00 a.m.: Hb 11 g/dL, RBC 4.37 × 10^12^/L, PLT 283 × 10^9^/L, PT/, APTT 35 s, INR 3, Fibrinogen 724 mg/dL, Antithrombin/. **20 September 2014** At 07:00 a.m.: Hb 10.4 g/dL, RBC 4.08 × 10^12^/L, PLT 325 × 10^9^/L, PT/, APTT/, INR/, Fibrinogen/, Antithrombin/. **21 September 2014** At 09:05 a.m.: Hb 10.4 g/dL, RBC 4.08 × 10^12^/L, PLT 325 × 10^9^/L. At 09:33 a.m.: INR 4.7, Fibrinogen 948 mg/dL, D-dimer 978 ng/mL. At 09:40 p.m.: Hb 8.9 g/dL, RBC 3.46 × 10^12^/L, PLT 328 × 10^9^/L, PT uncoagulable, APTT uncoagulable, INR uncoagulable, Fibrinogen uncoagulable, Antithrombin 88%, D-dimer 929 ng/mL.	**11 September 2014** Bemiparin sodium (7500 units, 1 vial) **12 September 2014** (Bemiparin sodium, 7500 units, 1 vial) Warfarin (2 tablets) **14 September 2014** Bemiparin sodium (7500 units, 1 vial) **18 September 2014** Warfarin ½ tablet **19 September 2014** Warfarin 1 tablet **20 September 2014** Warfarin ½ tablet **21 September 2014** Vitamin K 2 vials Prothrombin complex concentrate (28,000 IU + 2800 IU)
Case 10	85	Parkinson’s disease, previous bilateral hip implants, right periprosthetic fracture, subcapitate humerus fracture and wrist fracture.	Displaced femur fracture on the left coxofemoral prosthesis.	**19 November 2014**—05:37 p.m.: Admitted to the ED due to a complex fracture of the left femur on the coxofemoral prosthesis resulting from an accidental fall. 06:43 p.m.: Transferred to the Orthopedics department for a complex fracture of the left femur on the coxofemoral prosthesis. **21 November 2014**—08:30 a.m.: Exhibits worsening dyspnea, tachypnea, and hyperpyrexia. 08:44 a.m.: Arterial blood gas analysis reveals acute type 1 respiratory failure. Differently reduced respiratory rate in the chest, with prolonged expiratory phase and expiratory wheezing; bibasilar crackles. 09:15 a.m.: ECG shows sinus tachycardia and lateral right ventricular abnormalities. 09:50 a.m.: An anesthesiology consultation suggests deferring the surgery until the patient’s clinical condition stabilizes. 10:00 a.m.: Undergoes urological examination, revealing that the patient has dislodged the catheter, and attempts at re-catheterization were unsuccessful. 10:45 a.m.: Transferred to the ICU for acute respiratory failure (during the surgical procedure for the left femoral fracture). 01:00 p.m.: Decreased blood pressure, ++ hematuria (the catheter was dislodged in orthopedics and repositioned by the urologist). Chest CT scan shows no signs of pulmonary embolism in the main branches or segmental branches, no parenchymal lesions, or pleural effusion. 02:00 p.m.: Slight rise in temperature with blood presence in the urinary catheter. Hematuria. Hemodynamically stable, very drowsy, difficult to awaken. **22 November 2014**—Blood clots and blood in the urine, bleeding from the urethral meatus. At 13:00, taken to the operating room for left femoral skeletal traction. Urinary catheter changed, replaced with a 3-way catheter. Returns from the operating room at 13:30. Stable parameters, feverish, bladder washout for hematuria, nasogastric tube (NGT) removed. **23 November 2014**—08:00 a.m.: Stable parameters, bladder washout in progress. 10:27 a.m.: Cardiology consultation reveals increased troponin I levels (I 0.04; II 0.15; III 2.01 on 22.11). ECG from 21/11/14 shows ST segment depression from V4 to V6, different from the initial ECG; current ECG shows normal sinus rhythm and T wave in V4, with decreased ST segment depression. Normal echocardiogram with normal ventricular function (ejection fraction 50%) and mild aortic insufficiency. Bladder washout performed. 06:30 p.m.: Orthopedic consultation indicates poor general condition and high surgical risk. Only fracture stabilization is deemed necessary. Patient is highly agitated and has dislodged a peripheral needle; continuous bladder washout for hematuria. **24 November 2014**—04:30 p.m.: Patient returns from Orthopedic operating room, where they underwent surgery under general anesthesia for osteosynthesis of the periprosthetic fracture on the left side using a plate and screws. Arrives in the Recovery Room, drowsy, arousable to verbal and tactile stimuli. Conscious. Has cardiovascular system, blood system … from surgical wound. 06:00 p.m.: Drowsy, arousable to verbal and tactile stimuli, hypotensive, with unmeasurable urine output. Alb 5% 25 is administered. 08:00 p.m.: Patient unresponsive to verbal and tactile stimuli. Hemoglobin (Hb) 9.1. Transfusion of 2 units of packed red blood cells (PRBCs) performed. Coagulation test shows uncoagulable blood. Blood loss from the surgical wound. Arterial blood gas (ABG) analysis: pH 7.53, pCO2 27, pO2 119, K+ (potassium) levels, lactate 1.9. Operated limb highly edematous and bleeding. Patient in a state of shock, orotracheal intubation and ventilation, cardiopulmonary resuscitation (CPR) performed. Patient received reinfusion until 08:00 p.m., then a Redon drain was placed (the reservoir contained 150 mL of blood). At 08:00 p.m., the patient is unresponsive to stimuli. Significant blood loss from the surgical wound. Blood transfusion performed. 09:05 p.m.: Patient’s death is confirmed.	**19 November 2014** At 17:57 p.m.: HB 13.4 g/dL, RBC 4.21 × 10^12^/L, PLT 119 × 10^9^/L. **22 November 2014** At 07:00 a.m.: HB 9.3 g/dL, RBC 2.87 × 10^12^/L, PLT 90 × 10^9^/L, PT 84%, APTT 24 s, INR 1.1, Fibrinogen 608 mg/dL, Antithrombin/. **23 November 2014** At 07:00 a.m.: HB 8.6 g/dL, RBC 2.84 × 10^12^/L, PLT 97 × 10^9^/L, PT 102%, APTT 26 s, INR 1, Fibrinogen 638 mg/dL, Antithrombin/. At 17:43 p.m.: HB 8.8 g/dL, RBC 2.88 × 10^12^/L, PLT 92 × 10^9^/L. **24 November 2014** At 07:00 a.m.: HB 10.4 g/dL, RBC 3.34 × 10^12^/L, PLT 94 × 10^9^/L. At 17:42 p.m.: HB 9.1 g/dL, RBC 2.92 × 10^12^/L, PLT 108 × 10^9^/L.	**19 November 2014** Bemiparin sodium 3500 **21 November 2014** Packed RBCs 1 U Bemiparin sodium 3500 **22 November 2014** Bemiparin sodium 3500 **23 November 2014** Packed red blood cells 1 U + 1 U Bemiparin sodium 3500 **24 November 2014** Bemiparin sodium 3500 Packed RBCs 1 U
Case 11	59	Ex-smoker, type 2 diabetes mellitus, hypertension, obesity, dyslipidemia, previous NSTEMI.	Respiratory failure.	**12 January 2014**—At 11:21 p.m.: Admitted to the ED due to recent respiratory failure with fever and altered level of consciousness (temperature 36.5, FiO_2_ 5, SpO_2_ 76, SpO_2_ on O_2_ 90). **13 January 2014**—At 02:30 a.m.: Transferred to the ICU of Piombino Hospital for respiratory failure. Patient intubated and monitored, and insertion of NGT and triple-lumen CVC performed. **14 January 2014**—At 05:00 a.m.: Active bleeding observed from the mouth, CVC, and urinary catheter. Morning: Persistent nasal bleeding and bleeding from injection sites reported in the nursing diary. Afternoon: Severe clinical condition noted in the medical and nursing diaries, with hypotension and profuse bleeding from the oral cavity, nostrils, arms, and hematuria. Night: Bleeding from the airways and upper limbs reported in the nursing diary. **15 January 2014**—Morning: The medical and nursing diaries report bleeding with clots in the upper limbs. At 02:00 p.m.: Dressing changes performed on the upper limbs. Night: The medical and nursing diaries note additional bleeding from the upper limbs and airways. **16 January 2014**—Morning: Severe condition reported in the nursing diary with reduced bleeding. Sample collected for culture examination from the left arm clots, disinfected, and covered with sterile dressings. The medical diary reports bleeding from the entry sites of devices (CVC and urinary catheter). Afternoon: Very severe general condition noted in the medical and nursing diaries. Upper limbs dressed again. Cranial, thoracic, and abdominal CT scan performed, which does not show active bleeding. Night: Stable parameters with fever reported in the nursing diary. Sample collected for culture examination from the body fluid, resulting in positivity for Staphylococcus epidermidis. **17 January 2014**—Morning: Repeated blood aspiration and repositioning of the SNG reported in the nursing diary. Afternoon: Dressing changes on the upper limbs and as-needed blood aspiration reported in the nursing diary. Night: Stable parameters with fever reported in the medical and nursing diaries. **18 January 2014**—Stable parameters reported in the medical and nursing diaries with an episode of desaturation and tracheal tube obstruction due to the presence of blood encrustations along the tube wall and tip. Blood culture performed from the CVC. **19 January 2014**—Stable parameters, hyperpyrexia, and as-needed blood aspiration reported in the medical and nursing diaries. Dressing changes on the upper limbs. Blood sample collected for culture examination, resulting in positivity for Staphylococcus epidermidis. At 07:20 p.m.: Patient’s demise confirmed.	**12 January 2014** At 11:50 p.m.: HB 14.2 g/dL, RBC 5.32 × 10^12^/L, PLT 276 × 10^9^/L, PT/, APTT 27 s, INR 1.1. **14 January 2014** At 07:12 a.m.: HB 11.3 g/dL, RBC 4.31 × 10^12^/L, PLT 233 × 10^9^/L, Fibrinogen 535 mg/dL. At 01:02 p.m.: PT uncoagulable, APTT uncoagulable, INR uncoagulable, Fibrinogen uncoagulable. At 09:29 p.m.: HB 7.8 g/dL, RBC 2.9 × 10^12^/L, PLT 238 × 10^9^/L, PT 46%, APTT uncoagulable, INR 1.7, Fibrinogen uncoagulable, Antithrombin 70%, D-dimer 200 ng/mL. **15 January 2014** At 07:23 a.m.: HB 7.8 g/dL, RBC 2.9 × 10^12^/L, PLT 220 × 10^9^/L, PT 76%, APTT 34 s, INR 1.2, Fibrinogen 574 mg/dL, Antithrombin/, D-dimer 229 ng/mL. **19 January 2014** At 07:35 a.m.: HB 11.2 g/dL, RBC 4.24 × 10^12^/L, PLT 250 × 10^9^/L, PT 83%, APTT 26 s, INR 1.2, Fibrinogen 648 mg/dL, Antithrombin/, D-dimer/.	**13 January 2014** Fondaparinux sodium 2.5 mg Enoxaparin sodium 8000 IU **14 January 2014** Vitamin K 2 vials Plasma 3 U Packed RBCs 2 units **15 January 2014** Packed RBCs 2 units **16 January 2014** Packed RBCs 1 unit
Case 12	84	Arterial hypertension, chronic atrial fibrillation, CABG, pacemaker carrier, TEA surgery, peripheral vasculopathy with amputation of the 4th and 5th fingers of the right foot, home therapy with Warfarin, ASA, and LMWH for approximately 1 week.	Chest pain in chronic anemia.	**25 December 2014**—10:28 p.m.: Admitted to the ED for suspected chronic anemia. Elevated cardiac enzymes reported on admission (troponin I 1.92 ng/mL, BNP 708 pg/mL) with reported chest pain. 00:05 a.m.: Transferred to the ICU due to troponin movement and anemia. 08:00 a.m.: Administered Clexane 6000 1f. Dyspneic and restless; SNG inserted due to the presence of bright red blood. Fecal occult blood test conducted. Urgent EGD recommended due to the presence of melena. 12:00 p.m.: Troponin I level of 3.21 ng/mL and CK-Mb mass of 10.8 ng/mL obtained. 12:15 p.m.: Initiated blood transfusion with unit number 5109 (BP 65/35, HR 80). 01:55 p.m.: Initiated second unit with number 12617 (BP 140/76, HR 66). 02:20 p.m.: Initiated third unit with number 12267 (BP 133/68, HR 68, Temp 36). Stable parameters. 11:23 p.m.: Troponin I level obtained for the 24th hour, showing 2.82 ng/mL. **27 December 2014**—09:00 a.m.: Patient restless, no presence of blood observed from the nasogastric tube. 05:00 p.m.: Drowsy and unresponsive, undergoes a cranial CT scan that shows no intra- or extra-parenchymal hemorrhagic areas or focal lesions. 05:45 p.m.: Clexane reduced due to suspicion of a possible ischemic event. 08:00 p.m.: the presence of hematuria is reported. 09:00 p.m.: the presence of hematuria is reported. 00:25 a.m.: The patient’s passing is confirmed.	**25 December 2014** At 10:43 p.m.: HB 7.1 g/dL, RBC 2.34 × 10^12^/L, PLT 125 × 10^9^/L, PT 36%, APTT 34 s, INR 1.9, Fibrinogen/, Creatinine 2.04 mg/dL. **26 December 2014** At 07:00 a.m.: HB 6.7 g/dL, RBC 2.21 × 10^12^/L, PLT 119 × 10^9^/L, PT 37%, APTT 36 s, INR 1.9, Fibrinogen 409 mg/dL, Creatinine 2.06 mg/dL. At 05:17 p.m.: HB 9.8 g/dL, RBC 3.17 × 10^12^/L, PLT 111 × 10^9^/L. **27 December 2014** At 07:01 a.m.: PT 41%, APTT 40 s, INR 1.8, Fibrinogen/, Creatinine 1.93 mg/dL. At 04:15 p.m.: HB 9.6 g/dL, RBC 3.14 × 10^12^/L, PLT 112 × 10^9^/L. At 08:51 p.m.: HB 10.2 g/dL, RBC 3.34 × 10^12^/L, PLT 109 × 10^9^/L, Fibrinogen repeat.	**26 December 2014** Enoxaparin sodium (6000 IU 2 vials) Packed RBCs (1 unit + 1 unit + 1 unit) **27 December 2014** Enocxaparin sodium (6000 IU 1 vial + 2000 IU 1 vial)
Case 13	87	Right femur fracture.	Recent percutaneous aortic valvuloplasty surgery.	**30 July 2015**—Discharged from the Cardiology Department to undergo percutaneous aortic valvuloplasty at another hospital. The patient presented to the ED with a right femur fracture after an accidental fall. She was on Coumadin therapy (INR 2.1), which was temporarily suspended, and received half a dose of Konakion intravenously (resulting in INR 2.6). **3 August 2015**—04:40 p.m.: The aforementioned procedure was performed, and the patient returned to the hospital in the UCU. Physical examination: mild crackles. Cardiac auscultation: regular heart action. Patient reports itching. Sinus rhythm. Mild pain at admission. **4 August 2015**—Stable vital signs. Spontaneous breathing on room air. Lung examination: some crackles. Cardiac examination: systolic murmur 2/6 remains unchanged. Wound on the femur dressed twice with positive results. Slightly febrile. **5 August 2015**—Restless patient with general malaise. ECG: incomplete left bundle branch block with marked left axis deviation. Stationary condition. Anxious. Physical examination unchanged. Disoriented. Abundant diuresis. Slightly febrile. **6 August 2015**—Psychiatric and physical therapy consultation. Stable vital signs. **7 August 2015**—ECG: unchanged. Stationary condition. Experiencing back pain. Lung examination: crackles at the right base. Cardiac examination: unchanged. Calm patient. 07:00 p.m.: Developed fever with chills, administered intravenous antibiotics + cortisone. The patient never regained consciousness and repeatedly moved her legs out of bed. **8 August 2015**—Stable condition. Mild fever. Attempted ambulation, but patient non-compliant. Developed fever with chills, administered Rocefin. Physical examination unchanged. Rested throughout the night. **9 August 2015**—Stationary condition. Afebrile. Wound in good condition. Sutures removed. ECG: Non-specific abnormalities in ventricular repolarization. Clinical condition stable. Right hip pain. Restless patient. Decreased heart rate and electromechanical dissociation. Intubation performed along with resuscitation maneuvers and administration of three doses of adrenaline, resulting in the return of heart rate and blood pressure. Subsequent ventricular fibrillation occurred, followed by DC shock without success. Cordarone and adrenaline administered. 08:30 p.m.: Patient pronounced deceased.	**4 August 2015** At 07:00 a.m.: HB 9.5 g/dL, RBC 3.29 × 10^12^/L, PLT 135 × 10^9^/L, PT 100%, APTT 30 s, INR 1, Fibrinogen 479 mg/dL, Creatinine 0.58 mg/dL. **5 August 2015** At 07:00 a.m.: HB 10.1 g/dL, RBC 3.43 × 10^12^/L, PLT 132 × 10^9^/L, Creatinine 0.48 mg/dL. **6 August 2015** At 07:00 a.m.: HB 9.4 g/dL, RBC 3.25 × 10^12^/L, PLT 149 × 10^9^/L. **7 August 2015** At 07:01 a.m.: HB 9.6 g/dL, RBC 3.37 × 10^12^/L, PLT 271 × 10^9^/L.	**3 August 2015** Enoxaparin sodium (400 IU, 1 vial) **4 August 2015** Enoxaparin sodium (400 IU, 1 vial) **5 August 2015** Enoxaparin sodium (400 IU, 1 vial) **6 August 2015** Enoxaparin sodium (400 IU, 1 vial) **7 August 2015** Enoxaparin sodium (400 IU, 1 vial) **8 August 2015** Enoxaparin sodium (400 IU, 1 vial)
Case 14	90	Syncope and bradycardia in permanent atrial fibrillation under home therapy with Coumadin; COPD; aortic valve. Previous TIA and TEA procedure. Previous radiotherapy for epithelioma.	Cardiac arrhythmia.	**11 January 2015**—At 11:51 p.m.: Admitted to the ED due to a syncopal episode with a laceration on the scalp. The patient presents bradycardia and hypotension (heart rate 44, blood pressure 65/90). A cranial CT scan is performed, which does not reveal focal brain or bone lesions. **12 January 2015**—At 01:00 p.m.: Transferred to the Coronary Care Unit for rhythm disturbances and lightheadedness. The clinical condition is stable. An ECG shows normofrequent atrial fibrillation, low voltages in peripheral leads, and right bundle branch block, and an echocardiogram reveals moderate mitral insufficiency, concentric hypertrophy, moderate tricuspid insufficiency, and no pericardial effusion. At 11:50 p.m.: Blood tests show a troponin I level of 5.21 ng/mL. **13 January 2015**—At 09:00 a.m.: An ECG shows “Right bundle branch block and diffuse T-wave abnormalities”, with increasing troponin I levels. A bedside chest X-ray shows “interstitial congestive involvement and cardiomegaly”. **14 January 2015**—At 07:00 a.m.: An ECG shows “Normofrequent atrial fibrillation, right bundle branch block, sporadic ventricular extrasystole, signs of left ventricular overload”. At 10:40 a.m.: The medical record states “Restart Coumadin therapy…” At 11:50 a.m.: Sudden hypotension and bradyarrhythmia (33 bpm) with loss of consciousness occur, treated with Trendelenburg position and administration of dopamine in 250 cc of saline solution and Hydrocortisone 1000 mg. The medical record notes “Unconscious patient, after MCR patient regained consciousness and increased blood pressure … patient in a critical condition”. At 02:30 p.m.: The patient is pronounced deceased.	**11 January 2015** At 11:51 p.m.: HB 12.4 g/dL, RBC 3.99 × 10^12^/L, PLT 100 × 10^9^/L, PT/, APTT 36 s, INR 2.4, Creatinine 1.43 mg/dL. **12 January 2015** At 07:01 a.m.: HB 12.1 g/dL, RBC 3.96 × 10^12^/L, PLT 102 × 10^9^/L, INR 2.6, Creatinine 1.40 mg/dL.	**12 January 2015** Saline solution (500 mL) Oxygen therapy (1 L/min with nasal cannula) Theophylline (1 mL IV) Omeprazole, Normal saline 500 mL + Aminophylline, Dutasteride, and Enoxaparin (0.4 mL) **13 January 2015** Irbesartan (300 mg half tablet) Furosemide (25 mg) Silodosin Enoxaparin sodium (0.4 mL), Dutasteride **14 January 2015** Bisoprolol (1.25 mg) Irbesartan (150 mg, 1 tablet) Silodosin Dopamine, Hydrocortisone (1000 mg)

ABG: arterial blood gas; AF: atrial fibrillation; APTT: Activated Partial Thromboplastin Time; ARF: acute respiratory failure; ASA: aspirin; ATIII: antithrombin III; BNP: brain natriuretic peptide; BP: blood pressure; CABG: coronary artery bypass graft; CF: cardiac frequency; CSH: chronic subdural hematoma; CVC: central venous catheter; DIC: disseminated intravascular coagulation; ECG: electrocardiography; ED: Emergency Department; EF: ejection fraction; EGD: esophagogastroduodenoscopy; ENT: ear, nose, throat; HB: hemoglobin; HBV: Human Hepatitis B Virus; HCV: Human Hepatitis C Virus; HF: heart failure; ICU: Intensive Care Unit; INR: International Normalized Ratio; LMWH: low-molecular-weight heparin; NSTEMI: non-ST-elevation myocardial infarct; PE: pulmonary embolism; PEA: pulseless electrical activity; PEG: Percutaneous Endoscopic Gastrostomy; OR: operative room; PLT: platelets; PM: post mortem; PT: prothrombin time; RBC: red blood cells; CRP: C reactive protein; TEA: Transcatheter Edge-to-Edge Repair; TIA: transient ischemic attack.

**Table 2 diagnostics-13-03361-t002:** Histopathological investigation results.

Cases	Brain	Lungs	Heart	Liver	Spleen	Kidneys	Other
Case 1	Advanced autolithic alterations	Advanced autolithic alterations, endoalveolar edema	Advanced autolithic alterations, arteriolosclerosis	Advanced autolithic alterations	Advanced autolithic alterations	Advanced autolithic alterations	
Case 2	Advanced autolithic alterations	Autolithic alterations, endoalveolar edema and hemorrhages	Putrefactive alteration	Putrefactive alteration	Putrefactive alteration	Putrefactive alteration	
Case 3	Putrefactive alteration	Putrefactive alteration, slight endoalveolar edema	Advanced putrefactive alteration, small interstitial hemorrhages	Putrefactive alteration	Putrefactive alteration	Advanced putrefactive alteration	Skin from thighs: putrefactive alteration, small interstitial hemorrhages. Glycophorin A +
Case 4	Advanced autolithic alterations	Advanced autolithic alterations, endoalveolar edema and erythrocytes	Advanced autolithic alterations	Autolithic alterations		Autolithic phenomena	Skin and muscle tissue from left thigh: putrefactive alteration, small interstitial hemorrhages. Glycophorin A +
Case 5	Advanced autolithic alterations, perivascular edema. Positive reaction to Glycophorin A in encephalic structures	Advanced putrefactive alteration, endoalveolar edema	Autolithic alterations, arteriolosclerosis	Advanced autolithic alterations	Advanced autolithic alterations	Autolithic alterations, arteriolosclerosis	Gastric wall: autolithic alterations Small intestine wall: autolithic alterations Skin from dorsal surface on right elbow: putrefactive alteration, small interstitial hemorrhages. Glycophorin A + Skin from dorsal surface on right wrist: putrefactive alteration, small interstitial hemorrhages. Glycophorin A + Skin from medial surface of left ankle: putrefactive alteration, small interstitial hemorrhages. Glycophorin A +
Case 6	Putrefactive alteration, perivascular and perineuronal edema	Putrefactive alteration, edema, focal atelectasis, microembolization aspects	Connective substitution, coronarosclerosis, putrefactive alteration	Putrefactive alteration	Putrefactive alteration	Advanced putrefactive alteration	Skin and muscle tissue from VC insertion site: putrefactive alteration, small interstitial hemorrhages. Glycophorin A + Skin from dorsal surface of right forearm: putrefactive alteration, small interstitial hemorrhages. Glycophorin A + Skin and muscle tissue from left thigh: putrefactive alteration, small interstitial hemorrhages. Glycophorin A +
Case 10	Advanced autolithic alterations	Advanced autolithic alterations, endoalveolar edema	Advanced autolithic alterations, arteriolosclerosis	Advanced autolithic alterations	Advanced autolithic alterations	Advanced autolithic alterations	
Case 14	Autolithic alterations, arteriolosclerosis	Putrefactive phenomena	Autolithic alterations	Advanced autolithic alterations	Autolithic alterations, milza senile	Autolithic alterations	

**Table 3 diagnostics-13-03361-t003:** Results of blood analysis conducted on patients during hospitalization.

Cases	aPTT	PT	TT	INR	RT	Heparin Blood Concentration	High Heparin Blood Concentration/Heparin Administration Compatibility
Case 1	Uncoagulable	Uncoagulable	Severe prolongation	Normal	Normal	>2 UI/ml	Yes
Case 2	Severe prolongation	Severe prolongation	Uncoagulable	Pharmacologically induced mild elevation	Normal	7.4 UI/ml	Yes
Case 3	Uncoagulable	Uncoagulable	Uncoagulable	Pharmacologically induced mild elevation	Normal	>2 UI/ml	Yes
Case 4	Severe prolongation	Uncoagulable	Uncoagulable	Normal	Normal	>2 UI/ml	Yes
Case 5	Uncoagulable			Pharmacologically induced moderate elevation			Yes
Case 6	Uncoagulable	Uncoagulable	Uncoagulable	Normal	Normal		Yes
Case 7	Uncoagulable	Uncoagulable					Yes
Case 8	Uncoagulable	Uncoagulable		Pharmacologically induced moderate elevation			Yes

aPTT: Activated Partial Thromboplastin Time; PT: Prothrombin Time; TT: Thrombin Time; INR: International Normalized Ratio; RT: Reptilase Time.

**Table 4 diagnostics-13-03361-t004:** Summary of clinical characteristics of cases exposed to anticoagulant administration and reconstruction of the causal link between anticoagulant administration and death.

	Cases	Significant Hemorrhagic Events	Causal Link between Hemorrhagic Event and Death	Heparin-Specific Laboratory Abnormalities	Heparinemia Dosage in Life
Group I	Case 1	Bleeding from surgical wound (post-femoral osteosynthesis surgery)	+	+	+
	Case 2	Bleeding after PEG placement and tracheostomy	+	+	+
	Case 3	Hematuria, hepatic artery thrombosis, and splenic infarction	+	+	+
	Case 4	Bleeding from NGT, mouth, and hematoma in the femoral area	+	+	+
Group II	Case 5	Subdural hematoma	+	+	-
	Case 6	Bleeding from venous access sites and NG tube	+	+	-
	Case 7	Bleeding from surgical wound (post-humeral osteosynthesis surgery)	+	+	-
	Case 8	Nasal–oral bleeding, DIC	+	+	-
	Case 9	Hematuria, bleeding from mouth and oro-tracheal tube	+	+	-
	Case 10	Hematuria and bleeding from surgical wound (post-femoral osteosynthesis surgery)	+	+	-
Group III	Case 11	Bleeding from mouth, nose, upper limbs, CVC, urinary catheter	-	-	-
	Case 12	Hematuria	-	-	-
Group IV	Case 13	Not reported	-	-	-
	Case 14	Not reported	-	-	-

+: present; -: absent.

## Data Availability

The data is included in a legal case in Italy and for this reason it is not available.

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
