# Peer review of "Fourteen Deaths from Suspected Heparin Overdose in an Italian Primary-Level Hospital"

_diagnostics, 2023, doi:10.3390/diagnostics13213361_

Round 1
Reviewer 1 Report
Comments and Suggestions for Authors
Dear Authors!
The manuscript is a high-quality paper about a rare murder series by heparin in a hospital setting.
The manuscript, however, could be further improved. My suggestions:
- the autopsy findings are entirely missing - they should be included for all eight cases underwent autopsies
- it should be good (illustrative) to include some histological pictures – especially those with glycophorin A positivity
- Glycophorin A is sometimes referred as glycophorin A (table 2), sometimes as glicoforin (line 262)
- the manuscripts does not mentions, why autopsy was not performed on the other six cases.
Author Response
The manuscript is a high-quality paper about a rare murder series by heparin in a hospital setting.
Thank you so much for your kindness appreciation.
The manuscript, however, could be further improved. My suggestions:
- the autopsy findings are entirely missing - they should be included for all eight cases underwent autopsies
We included the requested data.
- - it should be good (illustrative) to include some histological pictures – especially those with glycophorin A positivity
- We did it, thank you.
- - Glycophorin A is sometimes referred as glycophorin A (table 2), sometimes as glicoforin (line 262)
- Fixed.
- the manuscripts does not mentions, why autopsy was not performed on the other six cases. - We added these explanations in the paper.
Reviewer 2 Report
Comments and Suggestions for Authors
Please change the names of medications to the International nonproprietary name rather than the brand name. It required a search to determine each drug and product name.
The title is quite appropriate however unless I missed it I do not see any verification of homicide (which is alluded to in the introduction). Perhaps listing that the same staff were present at each case (one or more) as otherwise medical error tragically happens and at least in the country I practice in is handled with great sensitivity as the Healthcare provider(s) are also suffering. ?Why was criminal activity suspected. There are 100's of staff who work within the system.
There is potential for a paper prior to this one (when/why to suspect a homicide within the medical domain.
Author Response
Please change the names of medications to the International nonproprietary name rather than the brand name. It required a search to determine each drug and product name.
Thank you for your precious comments.
The title is quite appropriate however unless I missed it I do not see any verification of homicide (which is alluded to in the introduction). Perhaps listing that the same staff were present at each case (one or more) as otherwise medical error tragically happens and at least in the country I practice in is handled with great sensitivity as the Healthcare provider(s) are also suffering?
We modified the paper following your suggestions. The tables and the discussion were modified.
Why was criminal activity suspected. There are 100's of staff who work within the system.
We explained these points in tables and discussion.
Round 2
Reviewer 2 Report
Comments and Suggestions for Authors
Thank you for the edits. The links between medicine and the forensic investigation is much more clear. There are a few terms that are not used Internationally that could be substituted (ie Uman complex = prothrombin complex concentrate)